# Using Exogenous Melatonin, Glutathione, Proline, and Glycine Betaine Treatments to Combat Abiotic Stresses in Crops

**DOI:** 10.3390/ijms232112913

**Published:** 2022-10-26

**Authors:** Memoona Khalid, Hafiz Mamoon Rehman, Nisar Ahmed, Sehar Nawaz, Fozia Saleem, Shakeel Ahmad, Muhammad Uzair, Iqrar Ahmad Rana, Rana Muhammad Atif, Qamar U. Zaman, Hon-Ming Lam

**Affiliations:** 1Centre for Agricultural Biochemistry and Biotechnology (CABB), University of Agriculture Faisalabad, Faisalabad 38000, Pakistan; 2Center for Soybean Research of the Partner State Key Laboratory of Agrobiotechnology and School of Life Sciences, The Chinese University of Hong Kong, Shatin, Hong Kong; 3Seed Center, Ministry of Environment, Water & Agriculture, Riyadh 14712, Saudi Arabia; 4Department of Biochemistry & Cellular and Molecular Biology, University of Tennessee, Knoxville, TN 37996, USA; 5Center for Advanced Studies in Agriculture and Food Security, University of Agriculture Faisalabad Pakistan, Punjab 38000, Pakistan

**Keywords:** antioxidants, drought, heat, salinity, heavy metals, abiotic stress, melatonin, glutathione, proline, glycine betaine

## Abstract

Abiotic stresses, such as drought, salinity, heat, cold, and heavy metals, are associated with global climate change and hamper plant growth and development, affecting crop yields and quality. However, the negative effects of abiotic stresses can be mitigated through exogenous treatments using small biomolecules. For example, the foliar application of melatonin provides the following: it protects the photosynthetic apparatus; it increases the antioxidant defenses, osmoprotectant, and soluble sugar levels; it prevents tissue damage and reduces electrolyte leakage; it improves reactive oxygen species (ROS) scavenging; and it increases biomass, maintains the redox and ion homeostasis, and improves gaseous exchange. Glutathione spray upregulates the glyoxalase system, reduces methylglyoxal (MG) toxicity and oxidative stress, decreases hydrogen peroxide and malondialdehyde accumulation, improves the defense mechanisms, tissue repairs, and nitrogen fixation, and upregulates the phytochelatins. The exogenous application of proline enhances growth and other physiological characteristics, upregulates osmoprotection, protects the integrity of the plasma lemma, reduces lipid peroxidation, increases photosynthetic pigments, phenolic acids, flavonoids, and amino acids, and enhances stress tolerance, carbon fixation, and leaf nitrogen content. The foliar application of glycine betaine improves growth, upregulates osmoprotection and osmoregulation, increases relative water content, net photosynthetic rate, and catalase activity, decreases photorespiration, ion leakage, and lipid peroxidation, protects the oxygen-evolving complex, and prevents chlorosis. Chemical priming has various important advantages over transgenic technology as it is typically more affordable for farmers and safe for plants, people, and animals, while being considered environmentally acceptable. Chemical priming helps to improve the quality and quantity of the yield. This review summarizes and discusses how exogenous melatonin, glutathione, proline, and glycine betaine can help crops combat abiotic stresses.

## 1. Introduction

Plants, being sessile in nature, must withstand various abiotic stresses, including drought, salinity, extreme temperatures, and heavy metals, which they accomplish through physiological means rather than behavioral avoidance. These abiotic stresses have severe negative impacts on crop productivity, growth, and development. It is therefore of vital importance to improve plant physiological and biochemical functions by protecting their cellular environments under abiotic stresses to fulfill food security needs [1].

Drought is considered one of the most severe abiotic stresses that causes osmotic imbalance in plants; due to unpredictable rainfall, light intensity, and fluctuating temperatures, the electron transport rate, stomatal conductance, photosynthesis, and carbon dioxide fixation are reduced. Salinity stress is another major threat to crop productivity; exposure to high salinity causes both ionic and osmotic imbalances, resulting in the overproduction of reactive oxygen species (ROS). Extreme temperatures can be too high or too low for normal physiological functions in the plant, limiting agricultural production. Heat stress causes the denaturation of enzymes and protein structures. Cold stress is induced by freezing temperatures, which negatively affect plant growth and yield. Extreme temperatures lead to the overproduction of ROS, which causes lipid peroxidation, protein damage, or cell structure destruction. Heavy metal stress is also an increasing threat to agricultural land and crop productivity. The uptake of heavy metals and metalloids such as cadmium (Cd), lead (Pb), and arsenic (As) from contaminated soil results in overproduction of ROS, oxidative damage, and reduction in plant productivity [2] (Figure 1).

Antioxidants are the key elements that protect plants from the oxidative damage caused by abiotic stresses. Melatonin acts as an effective antioxidant against different abiotic stresses and enhances certain physiological activities, including osmotic regulation, growth, germination, photosynthesis, primary and secondary metabolism, anti-senescence, plant hormone regulation, transpiration, ROS scavenging, and antioxidant enzyme activities. For example, seed priming using melatonin enhanced the plant’s ability to scavenge ROS and improved the photosynthetic efficiency in various crops [3,4]. Glutathione is known as one of the most abundant and essential metabolites for combating abiotic stresses as an antioxidant. A study of the exogenous application of glutathione demonstrated an increase in the abiotic stress tolerance of plants by improving the growth parameters, chlorophyll, photosynthesis, and antioxidant enzyme activities [5]. Despite being an osmolyte, proline is also considered an antioxidant. The external application of proline during abiotic stress was shown to increase the growth, photosynthetic activity, endogenous proline content, and antioxidant enzyme activities, which protected the integrity of the plasmalemma, scavenged ROS, and stabilized proteins and other physiological functions of the plants [6]. Glycine betaine is an osmoprotectant that also works as an antioxidant against abiotic stresses. Exogenous glycine betaine played a significant role in the detoxification of ROS, increased photosynthesis and stomatal conductance, decreased photorespiration in certain crops, improved yield and relative membrane permeability, reduced stomatal closure and the accumulation of malondialdehyde and leaf damage, and alleviated Cd stress in various crops [7,8] (Figure 2).

Comparing chemical priming to transgenic technology, there are some substantial advantages that are affordable for farmers and safe for plants, humans, and animals while being inexpensive and environment friendly. The exogenous application of melatonin, glutathione, proline, and glycine betaine mitigate the adverse effects of abiotic stresses on plants and provide an alternative to the utilization of transgenic plants.

In this review, we will discuss the beneficial effects of externally applied melatonin, glutathione, proline, and glycine betaine, especially their antioxidant roles in crop plants under various abiotic stresses. 

## 2. Melatonin 

### 2.1. Structure and Function of Melatonin

Melatonin (N–acetyl-5-methoxytryptamine) is a widespread and highly conserved molecule across species that was initially discovered in 1995. It is an indolic compound derived from tryptophan with a very similar structure to indole-3-acetic acid (IAA); likewise, melatonin serves similar functions in metabolic pathways. It has an exceptionally diverse range of activities to the extent that it can be another plant hormone [9]. It acts as an antioxidant, neutralizing the overproduction of ROS under stressful conditions to reduce oxidative damage and improve the growth and development of plants. Various physiological processes are enhanced by melatonin to overcome abiotic stresses, such as stomatal conductance, photosynthesis, root dynamics, carbohydrate assimilation, and water relation. Transpiration, ROS scavenging, and antioxidant enzyme activities, which together serve to alleviate oxidative damage to lipids, proteins, and nucleic acids, are also enhanced by the application of melatonin. For example, seeds pre-soaked in melatonin had increased numbers and a wider opening of stomata, enhanced antioxidant enzyme activities, osmoprotectants, and ROS scavenging activity in many crops [3,10].

Seed germination and seedling development (the growth of roots and shoots) are two vital processes of a plant’s successful lifecycle. Melatonin has been observed to regulate these processes and increase plant stress tolerance. Plant biomass ultimately depends on the amount of assimilated CO_2_. Some processes are determined by the efficiency of the leaf photochemistry such as the light harvesting, chlorophyll production, photosynthetic electron transfer, and leaf gaseous exchange processes. It was reported that, under stress conditions, the external application of melatonin enhanced the photosynthetic rate in plants [9]. Melatonin is a master phytohormone regulator, especially for auxin levels, and it improves the ability of plants to tolerate stress conditions. Melatonin has been added to the current list of secondary messengers that control the operation of transporter proteins and maintain the ion homeostasis in plants [11]. Melatonin was found to regulate the dark reaction of photosynthesis through carbon fixation at the molecular level by upregulating the activity of various enzymes, such as ribulose bisphosphate carboxylase/oxygenase (RuBisCo), glyceraldehyde-3-phosphate dehydrogenase, phosphoglycerate kinase, phosphoribulokinase, and fructose-bisphosphate aldolase. Chlorophyll fluorescence was significantly increased by external melatonin application [12].

### 2.2. Melatonin Effect against Drought Stress 

Drought stress generally causes an osmotic imbalance in plants, which leads to the disruption of the membrane system, electron transport chain, stomatal conductance, photosynthesis, and carboxylation efficiency; these in turn lead to the over-production of ROS, which ultimately upsets the redox homeostasis, resulting in changes to the physiological and biochemical processes, particularly when the drought stress is prolonged and extreme. The exogenous application of melatonin has been proven to be beneficial against drought stress. It is effective at controlling drought-induced oxidative damage, leaf senescence, and inhibition of photosystems, ultimately improving the crop yield [13].

In *Arabidopsis thaliana* (Arabidopsis/Thale cress), adding 50 µM of melatonin supplement to the nutrient solution upregulated stress-responsive genes (*COR15A*, *RD22*, and *KIN1*) when grown under drought conditions, raising the level of soluble sugars such as sucrose when grown under drought conditions [14] (Table 1). Pretreating seeds with 100 µM of melatonin improved plant growth, osmoprotectant levels, induced stress-responsive genes, scavenged ROS, and reduced electrolyte leakage in *Oryza sativa* (Rice) when under drought stress [15]. In *Zea mays* (Maize), a 1 mM melatonin treatment upregulated photoprotection (increased photosystem PS Ⅱ efficiency), and a foliar application of 100 µM of melatonin increased stomatal conductance, photosynthesis and transpiration rates, cell turgor, water holding capacity, increased enzymatic and non-enzymatic antioxidants, regulated osmotic potential, and scavenged ROS to tolerate drought stress [16,17]. In *Triticum aestivum* (Wheat), the application of 500 µM of melatonin to the soil regulated photosynthesis, cell turgor, increased water holding capacity, scavenged ROS, and reduced membrane damage; seed pretreatment with 10 or 100 µM of melatonin (dependent on variety) increased germination percentage, increased radicle and plumule lengths, and increased lysine content (a germination-related amino acid) during drought stress [18,19]. The foliar application of 100 µM of melatonin to *Fagopyrum tataricum* (Tartary Buckwheat) increased osmoprotectants, water status, secondary metabolites, antioxidant enzymes, photosynthetic rate, and ROS scavenging when under drought stress [20]. The foliar or soil application of 1 mM of melatonin to *Hordeum vulgare* (Barley) increased the endogenous melatonin level, antioxidants, abscisic acid (ABA), water status, photosynthetic rate, and PS Ⅱ efficiency when grown under drought stress [21]. In *Glycine max* (Soybean), coating seeds with 50 µM of melatonin increased seedling biomass and seedling growth and reduced electrolyte leakage; the foliar or root application of 100 µM of melatonin increased plant growth, flowering, seed yield, gaseous exchange, PS Ⅱ efficiency, and antioxidant enzymes to combat drought stress [22,23]. In *Minhot esculenta* (Cassava), the application of 100 µM of melatonin to the soil increased peroxidase (POD) activity and scavenging of ROS to reduce the negative effects of drought stress [24]. In *Gossypium hirsutum* (Cotton) seeds pre-soaked with 100 µM of melatonin developed into plants with increased numbers, increased opening of stomata, enhanced antioxidant enzyme activities, osmoprotection, and ROS scavenging when grown under drought stress [25]. Soil treatment with 10 µM of melatonin in the case of *Medicago sativa* (Alfalfa) increased chlorophyll content, stomatal conductance, osmoprotection, upregulated nitro-oxidative homeostasis and ROS scavenging, and reduced cellular redox disruption against drought stress [26]. In *Malus domestica* (Apple), the application of 100 µM of melatonin to the soil increased the water holding capacity, rate of photosynthesis, stomatal regulation, and antioxidants; furthermore, it decreased the electrolyte leakage, ROS production, oxidative damage, and leaf senescence, which helped to avoid the consequences drought stress [27]. Roots of *Vitis vinifera* (Grape) were pretreated with 100 µM of melatonin and had increased photoprotection, leaf thickness, stomata size, enzymatic and non-enzymatic antioxidants, and reduced oxidative damage when grown under drought stress [4]. *Actinidia chinensis* (Kiwifruit) treated with 100 µM of melatonin had increased osmoprotectants, protein biosynthesis, photosynthesis, and reduced cell membrane damage against drought stress [12]. The foliar application of 100 µM of melatonin to *Carya cathayensis* (Chinese hickory) increased photosynthesis, antioxidants, osmoprotectants, and scavenging of ROS under drought stress [13]. The foliar application of 200 µM of melatonin to *Solanum lycopersicum* (Tomato) increased chlorophyll and antioxidant enzyme contents; a 0.1 mM melatonin application increased photosynthesis, PS Ⅱ efficiency, antioxidants, and reduced toxic substance contents to tolerate drought stress [28,29]. Seeds of *Capsicum annuum* (Pepper) were pretreated with 50 µM of melatonin and had an increased water holding capacity, endogenous melatonin, increased carotenoids, and increased chlorophyll contents to combat drought stress [30]. In *Citrullus lanatus* (Watermelon), a 150 µM melatonin pretreatment of the root increased wax accumulation and reduced the ABA level under drought stress [19]. In *Cucumis sativus* (Cucumber), seeds primed with a nutrient solution supplemented by 100 µM of melatonin had increased germination rates, root growths, chlorophyll, photosynthesis, antioxidant enzymes, and ROS scavenging, whereas a foliar application of 10 µM of melatonin resulted in ROS scavenging and increased drought tolerance [4,31]. In *Brassica napus* (Rapeseed), seed primed with 500 µM of melatonin had increased chlorophyll, stomatal regulation, cell wall expansion, antioxidant enzymes and osmoprotectants, which operated in tandem with a reduction in oxidative injury to combat drought stress [32]. Foliar applications of 100 µM of melatonin increased photosynthesis, chlorophyll, and osmoprotectants while reducing cell membrane damage and relative conductivity in *Dendranthema morifolium* (Jinyu Chuju) in order to avoid the consequences of drought stress [33]. In *Dracocephalum moldavica* (Moldavian balm), foliar applications of 100 µM of melatonin increased plant growth, flowering, antioxidant activities, chlorophyll contents, water holding capacity, and ROS scavenging when grown under drought stress [34]. Foliar applications of 20 µM of melatonin increased photosynthetic contents, water holding capacity, PS Ⅱ efficiency, and ROS scavenging while also reducing leaf senescence in *Agrostis stolonifera* (Creeping bentgrass) to combat drought stress [35]. In *Festuca arundinacea* (Tall fescue), an irrigation pretreatment with 20 µM of melatonin increased antioxidant enzyme activities, chlorophyll content, plant growth, and ROS scavenging under drought conditions [36]. In *Cynodon dactylon* (Bermuda grass), 20 and 100 µM of melatonin was included in irrigation pretreatments, which increased plant growth, chlorophyll content, and antioxidant activities while also inducing stress-responsive genes, improved hormonal regulation, and ROS scavenging to reduce the negative effects of drought stress [14]. Foliar applications of 100 and 300 µM of melatonin to *Trigonella foenum-graecum* (Fenugreek) increased the levels of endogenous melatonin, secondary metabolites, chlorophyll and antioxidant enzymes, and improved ROS scavenging under drought [37]. In *Coffea Arabica* (Coffee), the application of 300 µM of melatonin to soil increased photoprotection, gaseous exchange, carboxylation activities, chlorophyll content, and antioxidant enzyme activities to combat drought stress [38]. In *Camellia sinensis* (Tea), the foliar application of 100 µM of melatonin increased photosynthesis, antioxidant enzymes, and ROS scavenging while reducing glutathione (GSH) and ascorbic acid (AsA) contents when grown under drought stress [39]. In *Nicotiana benthamiana* (Tobacco), the foliar application of 10 µM of melatonin increased drought tolerance through ROS scavenging and reduced oxidative damage [31].

### 2.3. Melatonin Effect against Salinity Stress 

The negative impacts of salinity stress on agricultural production and plant growth include a reduction in the photosynthetic activity and disturbances in the protein and carbohydrate metabolisms. The first sign of salinity is the generation of ROS and their harmful physiological effects, such as protein degradation, lipid peroxidation, and DNA mutation, which result in oxidative damage and the downregulation of carbon dioxide fixation, leading to physiological dysfunction and programmed cell death. Salinity speeds up leaf senescence and reduces cell expansion and crop yield. The exogenous application of melatonin increases relative water content, antioxidant enzyme activity, and improves photosynthetic efficiency to enhance the tolerance against salinity stress [40,41]. 

Under high salinity conditions, priming *Momordica charantia* (Bitter melon) seeds with 150 µM of melatonin increased relative water content, antioxidant enzyme activities, stress-responsive gene expression levels, and decreased hydrogen peroxide and malondialdehyde levels [42] (Table 2). In *Zea mays* (Maize), the pretreatment of seeds with 0.4, 0.8, and 1.6 mM of melatonin improved the shoot and root lengths, rates of germination, fresh and dry weights of the seedlings, potassium ion (K^+^) contents, antioxidant enzyme activities, and relative water content under high salinity conditions [43]. In *Gossypium hirsutum* (Cotton), priming the seed with 25 µM of melatonin enhanced the plant’s ability to scavenge ROS and improved photosynthetic efficiency against salt stress [44]. Seed priming with 70 µM of melatonin enhanced photosynthetic pigments, indole-3-acetic acid (IAA) contents, and growth parameters in *Triticum aestivum* (Wheat) to avoid the consequences of salinity stress [45]. In *Ocimum basilicum* (Basil), seed priming with 10 µM of melatonin increased the contents of flavonoids and phenolic acids under high salinity conditions [46]. Priming *Vicia faba* (Faba bean) seeds with 100 or 500 mM of melatonin improved novel protein expressions against salt stress [47]. In *Cucumis sativum* (Cucumber), seed priming with 1 µM of melatonin enhanced seed germination under saline conditions [43]. The foliar treatment of *Arabidopsis thaliana* (Arabidopsis/Thale cress) using 10 µM of melatonin induced the antioxidant defense system, scavenged ROS, and upregulated abscisic acid (ABA) responsive genes under salinity stress [48]. The foliar application of *Brassica napus* (Rapeseed) using 1 µM of melatonin reduced lipid peroxidation and hydrogen peroxide content while also maintaining redox and ion homeostasis against salinity [49]. In *Brassica juncea* (Mustard greens), the foliar application of 1 µM of melatonin increased leaf length/width, plant height, and stem diameter, and improved gaseous exchange and relative water content; furthermore, it increased the salicylic acid level while reducing the abscisic acid (ABA) level against saline conditions [50]. Seed pretreatment of *Cucumis melo* (Melon) with 0, 10, and 50 µM of melatonin increased seed germination under high salinity [51]. In *Oryza sativa* (Rice), irrigating the roots with 0, 10, and 20 µM of melatonin upregulates antioxidants while inhibiting cell death and chlorophyll degradation against salt stress [52]. In a study on the foliar application of *Glycine max* (Soybean) using 0–100 µM of melatonin, the author noticed increased levels of photosynthesis, cell division, carbohydrates, fatty acids and ascorbate contents in addition to a reduction in the inhibitory effect of salt on gene expressions under high salinity conditions [22]. In *Malus hupehensis* (Pingyitiancha) seeds pretreated with 0.1 µM of melatonin, there was an increase in photosynthesis and ion homeostasis and a decrease in oxidative damage due to high salt contents [53]. In *Solanum lycopersicum* (Tomato), irrigating the roots with 100 µM of melatonin increased protein and membrane protection, antioxidants activities, and photosynthesis against salinity [54]. Seeds of *Citrullus lanatus* (Watermelon) that were pretreated using 50–150 µM of melatonin had increased antioxidant enzymes, photosynthesis, and PS Ⅱ efficiency in addition to reduced stomatal closure and oxidative damage while the plant was under salt stress [55].

### 2.4. Melatonin Effect against Heat Stress 

Heat can cause severe damage to plants. High temperatures affect plant growth, reducing the yield and affecting protein stability, which causes enzyme denaturation due to the overproduction of ROS. Heat stress disrupts the fluidity of the membrane, leading to a reduction in photosynthesis, growth, and yield. The exogenous application of melatonin reduces oxidative damage and upregulates heat shock factors to combat heat stress [56].

In *Triticum aestivum* (Wheat), the application of 20 µM of melatonin to soil application increased the rate of photosynthesis and reduced oxidative damage due to heat stress [56] (Table 3). The foliar application of melatonin on *Lolium perenne* (Perennial Ryegrass) regulated cytokinin biosynthetic genes, downregulated ABA biosynthetic genes, and enhanced the endogenous melatonin content against heat stress [57]. Seed pretreatment with 100 µM of melatonin enhanced phenolic acid levels, regulated transcript abundances, increased the endogenous melatonin level, and reduced oxidative stress in *Solanum lycopersicum* (Tomato) when under heat stress [58]. In *Arabidopsis thaliana* (Arabidopsis/Thale cress), the foliar application of 20 µM of melatonin upregulated heat shock factors to combat heat stress [59]. The application of 100 µM of melatonin to soil increased photosynthesis and reduced oxidative damage in *Zea mays* (Maize) grown under an elevated temperature [17].

### 2.5. Melatonin Effect against Cold Stress

Cold stress is an oxidative stress caused by chilling or freezing temperatures that can negatively affect plant growth and development. Cold stress induces the excessive generation of ROS, thus disrupting ROS homeostasis; this may lead to lipid peroxidation, cell structure damage, protein denaturation, and chilling injury. Cold stress reduces photosynthetic capacity, chloroplast development, and chlorophyll content. The exogenous application of melatonin improves antioxidant enzyme activities and reduces cold-induced oxidative stress [60].

In *Triticum aestivum* (Wheat), the exogenous application of 100 µM of melatonin improved antioxidant enzyme activities and reduced the oxidative stress caused by low temperatures [61] (Table 4). In *Citrullus lanatus* (Watermelon), the application of 150 µM of melatonin to soil increased the accumulation of hydrogen peroxide and increased cold stress tolerance [62]. In *Solanum lycopersicum* (Tomato), seed priming with 100 µM of melatonin improved photosynthesis and reduced oxidative damage due to cold stress [63]. The foliar application of 100 µM of melatonin increased the arabinose, mannose, and propanoic acid contents in *Cynodon dactylon* (Bermuda grass) when grown under cold stress [64]. In *Hordeum vulgare* (Barley), treatment with 1 mM of melatonin improved the water status, antioxidant system, and abscisic acid (ABA) level to tolerate cold stress [21]. In *Camellia sinensis* (Tea plant), the external application of melatonin increased antioxidant enzyme activities and reduced oxidative stress from exposure to low temperatures [65]. In *Oryza sativa* (Rice), exogenous melatonin improved antioxidant enzyme activities and reduced cold-induced oxidative stress [55]. In *Cucumis sativus* (Cucumber), the external application of 100 µM of melatonin improved antioxidant enzyme activities and reduced oxidative stress from low temperatures [66].

### 2.6. Melatonin Effect against Heavy Metal Stress

Heavy metals are either non-essential or minimally required elements for the normal growth and development of plants; they are found in the soil water of contaminated soils and are readily taken up by the plant, causing oxidative stress and damage to the plant due to the overproduction of ROS. The exogenous application of melatonin improves chlorophyll content, antioxidant enzymes, and ROS scavenging to combat heavy metal stress [67].

In *Triticum aestivum* (Wheat), the treatment of cadmium (Cd) heavy metal-contaminated soil with 50 µM of melatonin caused an increase in antioxidant enzymes [68] (Table 5). The foliar application of 50 µM of melatonin to *Medicago sativa* (Alfalfa) growing in heavy metal-containing soil resulted in an increase in ATP-binding cassette-containing (ABC) transporters and a decrease in cadmium (Cd) accumulation [69]. In *Solanum lycopersicum* (Tomato), seed pretreatment with 100 µM of melatonin increased antioxidants and plant growth and reduced electrolyte leakage and photoinhibition against cadmium (Cd) metal stress [21]. In *Nicotiana benthamiana* (Tobacco), a foliar application of 15 µM of melatonin increased cell growth and viability while decreasing DNA damage due to lead (Pb) heavy metal contents [53]. Soil treatment with 50 µM of melatonin increased antioxidants and plant biomass in *Cyphomandra betacea* (Tree tomato) to reduce the negative effects of cadmium (Cd) heavy metal stress [70]. In *Glycine max* (Soybean), seed priming with 100 mM of melatonin increased photosynthesis and antioxidants under aluminum (Al) heavy metal stress [53]. The foliar application of *Brassica oleracea* (Red cabbage) with 10 µM of melatonin increased the germination and fresh weight of the plant in the presence of copper (Cu) heavy metal [53]. In *Citrullus lanatus* (Watermelon), seed priming with 50 mg/L of melatonin increased plant growth, photosynthesis, chlorophyll content, antioxidant enzymes, and ROS scavenging to combat vanadium (V) heavy metal stress [71]. In *Zea mays* (Maize), treatment of heavy metal-containing soil with 500 µM of melatonin induced additional proteins related to stress reduction during germination [72]. In *Cucumis sativus* (Cucumber), irrigating cadmium (Cd) metal-containing soil with 100 or 150 µM of melatonin reduced the expressions of stress-inducible genes such as CsHA2 [43]. In *Amaranthus viridis* (Amaranthus), the foliar application of 400 µM of melatonin decreased the accumulation of lead (Pb) heavy metal in the roots [73].

## 3. Glutathione

### 3.1. Structure and Function of Glutathione

Glutathione (GSH; γ-glutamyl-cysteinyl-glycine), a low-molecular-weight thiol, is a very crucial metabolite involved in the plant’s antioxidant defense system. The external application of glutathione has been reported to increase growth parameters such as root and shoot lengths, fresh and dry weights, the number of fruits per plant, and fruit weight while enhancing the photosynthetic rate, chlorophyll a and b, and carotenoids; furthermore, glutathione increases antioxidants and osmoprotectants such as endogenous glutathione, total soluble sugars, proteins and polyamines and stimulates gene expressions (e.g., CaAPX1, CaMDHAR1, and CaDHAR1 in *Capsicum annuum*) and enzymatic activities, decreasing ROS production and the accumulation of malondialdehyde and other toxic substances [5]. 

It has been demonstrated that glutathione is a potent antioxidant that increases the plant’s resistance against abiotic stresses. Higher glutathione levels facilitate the detoxification of heavy metals, thus preventing any damage to lipids, amino acids, and polysaccharides, as well as any injury to the membranes, photosynthetic machinery, mitochondria, and other organelles, eventually conferring tolerance against abiotic stresses. Glutathione also plays a role in the maintenance of osmotic balance by stimulating osmoprotectants. Externally applied glutathione increases the endogenous glutathione level and improves the ratio of GSH (the reduced form) to GSSG (the oxidized form), enabling the efficient scavenging of excess ROS and therefore reducing oxidative stress and damage to plants. Methylglyoxal detoxification is one of the significant roles that the compound has in protecting the plant from damage by toxic substances. By interacting with other redox systems, glutathione directly or indirectly modulates the stress-responsive transcriptional and post-transcriptional levels [2].

### 3.2. Glutathione Effect against Drought Stress

Drought stress in plants causes leaf rolling, stunted plant growth, yellowing of leaves and leaf wilting by reduced leaf water potential and turgor pressure, increased stomatal closure, and decreased cell growth. Glutathione application increases plant stress tolerance by increasing antioxidants, abscisic acid, relative water content, and scavenging of ROS, thus improving growth parameters and chlorophyll contents. Exogenous application of glutathione improves chlorophyll content, photosynthesis, relative water content and antioxidant enzyme activities to combat drought stress [5].

In *Cicer arietinum* (Chickpea), seed soaking with 0.75 mM of glutathione increased the growth parameters, chlorophyll content, photosynthesis, endogenous proline level, and antioxidant enzyme activities under drought conditions [74] (Table 6). In *Oryza sativa* (Rice), spraying with 0.2 mM of glutathione increased the root and shoot lengths, dry and fresh weights, chlorophyll pigment, relative water content, and antioxidant enzyme activities when grown under drought [5]. The foliar application of glutathione to *Brassica napus* (Rapeseed) scavenged ROS and reduced oxidative damage due to drought stress [75]. In *Triticum aestivum* (Wheat)*,* treatment with exogenous glutathione via spraying improved the plant’s tolerance to drought, compared with the untreated cultivar [76]. In *Vigna radiata* (Mung bean)*,* the exogenous application of glutathione lessened drought-induced oxidative damage by enhancing the capacity of the antioxidant system and glyoxalase activity [77]. *Arabidopsis thaliana* (Arabidopsis/Thale cress) treated with glutathione via spraying had higher abscisic acid content, was more tolerant against drought stress, and had improved health under drought conditions [78].

### 3.3. Glutathione Effect against Salinity Stress

Salinity stress causes nutritional imbalance, inhibition of water uptake, photosynthesis and seed germination, and an overall decrease in crop productivity. Treatment with glutathione can remedy salinity stress by increasing plant growth, dry and fresh weights of roots and shoots, and total yield through the enhancement of water use efficiency, osmoprotectant levels, photosynthetic activities, and stomatal conductance. The exogenous application of glutathione increases water use efficiency, level of osmoprotectants, and antioxidant enzyme activity to combat salinity stress [59].

In *Capsicum frutescence* (Pepper) experiencing salinity stress, foliar sprays with 0.4 and 0.8 mM of glutathione increased water use efficiency, overall growth, fresh and dry weights of roots and shoots, yield, and the levels of osmoprotectants and antioxidants [79] (Table 7). In *Cucumis sativus* (Cucumber), soaking seeds in 0.5 mM of glutathione increased growth, fresh and dry weights, relative water content, photosynthetic activities, and stomatal conductance despite salinity stress conditions [80]. In the case of *Vicia faba* (Faba bean) under salt stress, foliar sprays with 0.5 mM of glutathione increased growth, fresh and dry weights, relative water content, photosynthetic activity, stomatal conductance, and antioxidant enzyme activities [81]. Foliar sprays with 1 mM of glutathione on *Triticum aestivum* (Wheat) counteracted the effects of salt stress by increasing plant growth, membrane stability, and the accumulation of osmoprotectants [59]. In the case of *Glycine max* (Soybean) under salt stress, foliar sprays with 1 mM of glutathione increased growth, photosynthesis, membrane stability, soluble sugar contents, and antioxidant enzyme activities [5]. Similarly, in *Phaseolus vulgaris* (Common bean), foliar sprays of 0.75 mM of glutathione increased the plant length, number and total area of leaves, fresh and dry weights of plant, relative water content, photosynthesis, and soluble sugars to avoid the consequences of salt stress [5]. *Arabidopsis thaliana* (Arabidopsis/Thale cress) treated with a glutathione foliar spray had higher abscisic acid levels, was more tolerant against salt stress, and had improved overall plant health under the stressful conditions [78]. In *Solanum lycopersicum* (Tomato), the exogenous application of glutathione reduced lipid peroxidation and improved the plant’s tolerance against salinity stress and oxidative stress [82]. In *Oryza sativa* (Rice), the spraying of glutathione improved the activities of antioxidant enzymes, reduced ROS accumulation, and reduced ROS-induced DNA damage due to salt stress [83]. 

### 3.4. Glutathione Effect against Heat Stress

Heat stress negatively impacts plants by reducing the leaf water potential and total leaf area, causing premature leaf senescence and thus affecting the growth and yield of the plant. Glutathione application increases resistance to heat by increasing antioxidant enzyme activities and the scavenging of ROS. The exogenous application of glutathione increases the protection of the plant against heat stress by maintaining relative water content and antioxidant enzyme activity [84].

The treatment of heat-stressed *Triticum aestivum* (Wheat) with glutathione increased the activities of antioxidant enzymes, therefore increasing the protection of the plant against heat stress [85] (Table 8). In *Vigna radiate* (Mung bean), pretreating the seeds with glutathione increased antioxidant enzyme activities and therefore enhanced heat stress resistance by decreasing the ROS level [77]. In *Cucumis sativus* (Cucumber), the external application of glutathione enhanced heat resistance, improved plant growth, chlorophyll content, and photosynthetic rate under heat stress [84]. In *Brassica campestris* (Mustard), treatment with glutathione maintained the relative water content and increased ROS scavenging and antioxidants during heat stress [86].

### 3.5. Glutathione Effect against Cold Stress

Cold stress delays leaf development in plants, prolongs the cell cycle with reduced cell production, stunts growth, and causes leaf chlorosis and a general disruption of the structure and functions of cells and tissues. However, external glutathione applications can reduce the adverse effects of cold stress and cause an increase in the lengths of roots and shoots and fresh and dry weights of the plant by increasing the endogenous glutathione level and decreasing electrolyte leakage and lipid peroxidation compared with untreated plants. The exogenous application of glutathione decreases electrolyte leakage and lipid peroxidation to combat cold stress [60].

In *Oryza sativa* (Rice), the spraying of 0.5 mM of glutathione increased the lengths of roots and shoots, as well as the fresh and dry weights and the endogenous glutathione level, compared with untreated plants under cold stress [88] (Table 9). In *Capsicum annum* (Pepper), the spraying of 0.5 mM of glutathione also increased the root and shoot lengths, fresh and dry weights, and endogenous glutathione level of plants subjected to cold stress [60]. The foliar application with glutathione in *Cucumis sativus* (Cucumber) decreased electrolyte leakage and lipid peroxidation when the plants were placed under cold stress [89]. In *Jatropha curcas* (Purging nut), the external application of glutathione enhanced the plant’s resistance to low temperatures and enhanced the activities of antioxidant enzymes [90].

### 3.6. Glutathione Effect against Heavy Metal Stress

Heavy metal stress generally causes toxic effects on plants, such as inhibition of growth, photosynthesis, and nutrient assimilation, along with altered water balance, chlorosis, and senescence. On the other hand, the application of glutathione has been observed to enhance stress tolerance in plants by increasing ROS scavenging, antioxidant activities, and levels of photosynthetic pigments. The exogenous application of glutathione increases photosynthetic pigments, alleviating oxidative damage to overcome the adverse effects of heavy metal stress [91].

In *Triticum aestivum* (Wheat), the foliar application of 20 µM of glutathione increased photosynthetic pigments and the endogenous level of glutathione in plants subjected to cadmium (Cd) heavy metal stress [62] (Table 10). In *Solanum melongena* (Brinjal), seed pretreatment with 1 mM of glutathione mitigated the adverse effects of arsenate (As) heavy metal stress with respect to protein damage [91]. The foliar application of 30 µM of glutathione to *Zea mays* (Maize) grown in cadmium (Cd) heavy metal-containing soil increased the secondary metabolites and flavonoids in the plants and alleviated oxidative damage [92]. In *Lolium multiflorum* (Italian ryegrass), the external application of 200 µM of glutathione increased lead (Pb) stress tolerance and the root and shoot biomass compared with untreated plants under the same heavy metal stress conditions [93]. In *Hordeum vulgare* (Barley), exogenous glutathione improved the antioxidant defense system and photosynthesis and decreased ROS accumulation due to cadmium (Cd) metal stress [94]. Exogenous glutathione applied to *Solanum lycopersicum* (Tomato) that was subjected to cadmium (Cd) metal stress synchronized the transcript levels of several stress-responsive transcription factors while also improving nitric oxide contents [95]. In *Oryza sativa* (Rice), treatment with glutathione elevated the endogenous glutathione level, mineral elements, and pigment contents, upregulated phytochelatins, and synchronized antioxidant enzyme activities during cadmium metal stress [96]. In *Brassica campestris* (Mustard), exogenous glutathione reduced the level of cadmium in roots and leaves as well as the accumulation of ROS, protecting the plant against heavy metal stress [97].

## 4. Proline

### 4.1. Structure and Function of Proline

Due to its cyclic structure and possession of a secondary amino group (the α-amino group) which differentiates it from the other proteinogenic amino acids, proline (Pro) is regarded as one of the most effective osmoprotectants and signaling molecules aside from its crucial role in the primary metabolism both as free amino acids and as a component of proteins. Numerous data sources point to a beneficial correlation between proline accumulation and enhanced plant stress tolerance. Proline is a significant component of the physiological response to stress in many plant species because it can accumulate in the cytosol without harming cellular structures. Due to its capacity to form hydrogen bonds, proline can increase protein stability and safeguard membrane integrity. In addition, proline can also protect cells by enhancing water uptake potential and boosting enzyme activation. Proline is an osmolyte and also a powerful antioxidant molecule, metal chelator, protein stabilizer, ROS scavenger, and inhibitor of programmed cell death [6].

External applications of proline can maintain the turgidity of the cell under stress, protect plants from harmful radiations, increase photosynthetic and transpiration rate, stomatal conductance, and antioxidants, thus increasing the growth and yield of plants even when under abiotic stresses. However, the excessive applications of proline can also impart toxic effects on plants [100].

### 4.2. Proline Effect against Drought Stress

The exogenous application of proline promotes the uptake of nutrients, increases the concentration of soluble proteins, and enhances the non-enzymatic antioxidant defense system to combat drought stress. In *Zea mays* (Maize), seed priming with 1 mM of proline increased photosynthetic and transpiration rates and stomatal conductance, while the foliar application of proline promoted the uptake and accumulation of nitrogen, phosphorus, and potassium, and enhanced the tolerance against drought stress [6] (Table 11). In *Triticum aestivum* (Wheat), the foliar application of 150 ppm of proline reduced the malondialdehyde level and lipid peroxidation caused by drought [101]. In *Chenopodium quinoa* (Quinoa), the foliar application of proline increased photosynthetic pigments, phenols, free amino acids, plant height, and the dry and fresh weights of roots and shoots compared with untreated plants under drought stress [102]. An external spray of proline in *Arabidopsis thaliana* (Arabidopsis/Thale cress) helped scavenge ROS and protected the integrity of the plasma lemma during drought stress [100]. In *Pisum sativum* (Pea), the foliar spray of 4 mM of proline increased the yield and the concentration of soluble proteins, enhancing the non-enzymatic antioxidant defense system compared with untreated plants under the same drought conditions [103].

### 4.3. Proline Effect against Salinity Stress

The exogenous application of proline increases chlorophyll, relative water content, transpiration rate, and stomatal conductance to alleviate salinity stress. In *Cucumis melo* (Muskmelon), a foliar spray of 10 mM of proline increased the growth, chlorophyll, endogenous proline, and relative water contents of plants experiencing salinity stress [104] (Table 12). In *Cucumis sativus* (Cucumber), supplementing the nutrient solution with 10 mM of proline increased the growth, endogenous proline content, and antioxidant enzyme activities despite saline conditions [105]. In *Glycine max* (Soybean), the external application of 25 mM of proline increased the growth, endogenous proline content, antioxidant enzyme activities, and nitrogen fixation compared with untreated plants under salinity stress conditions [106]. In *Helianthus annus* (Sunflower), foliar sprays of 30 and 60 mM of proline increased the growth, endogenous proline content, antioxidant enzyme activities, and free amino acid contents [107]. In *Zea mays* (Maize), foliar sprays with 30 mM of proline increased the growth and endogenous proline content under salt stress conditions [108]. *Triticum durum* (Durum wheat) seeds pretreated with 12 mM of proline saw increased growth, photosynthetic activities, endogenous proline content, and antioxidant enzyme activities in saline conditions [109]. A foliar spray with 30 mM of proline increased the growth, relative water content, gaseous exchange, free amino acids, and proline contents in *Sorghum bicolor* (Great millet) despite saline conditions [110]. In *Oryza sativa* (Rice), seeds pretreated with 1, 5, and 10 mM of proline increased the growth, seed germination, and chlorophyll and endogenous proline contents to mitigate the effects of salt stress [111]. In *Brassica juncea* (Mustard greens), the foliar application of 20 mM of proline improved the yield and salt stress tolerance [112]. The foliar application of 0.8 mM of proline to *Capsicum annum* (Red pepper) improved antioxidant enzyme activities, photosynthetic and transpiration rates, dry and fresh weights of the plant, and root and shoot lengths under salt stress conditions [113]. The foliar application of proline to *Daucus carota* (Wild carrot) improved antioxidant enzyme activities under salt stress as well as potassium and calcium levels in the roots and shoots [114]. In *Phaseolus vulgaris* (Bean), the foliar application of proline improved antioxidant enzymes activities and the endogenous proline level in response to salinity [115]. In *Raphanus sativus* (Radish), the foliar application of proline improved the transpiration rate, stomatal conductance, photosynthetic pigment contents, and contents of proteins and other nutrients to alleviate salinity stress [116]. The foliar application of proline to *Vicia faba* (Faba bean) experiencing salinity stress resulted in an improvement in the contents of photosynthetic pigments, soluble carbohydrates, endogenous proline, and free amino acids [6]. The foliar application of 5 µM of proline improved plant growth, photosynthetic rate, chlorophyll content, and yield in salt-stressed *Lactuca sativa* (Lettuce) [117].

### 4.4. Proline Effect against Heat Stress

In *Abelmoschus esculentus* (Okra), the foliar application of proline improved the shoot length, the number of leaves per plant, and free amino acid contents during heat stress [118]. Similarly, in *Lactuca sativa* (Lettuce), the foliar application of 5µM proline improved plant growth, photosynthetic rate, chlorophyll content and yield [117]. In *Vigna radiate* (Mung bean), the foliar application of proline improved CO_2_ assimilation capacity and heat tolerance [119] (Table 13).

### 4.5. Proline Effect against Cold Stress

The foliar application of proline increases antioxidant enzyme activities and the content of phenolic acids, flavonoids, and endogenous level of proline to overcome cold stress. In *Citrus reticulata* (Mandarin orange)*, Citrus sinensis* (Sweet orange)*,* and *Citrus paradise* (Grapefruit), foliar applications of proline increased the contents of phenolic acids, flavonoids, and endogenous proline and enhanced antioxidant enzyme activities caused by low temperatures [120]. In *Capsicum annum* (Red pepper), the foliar application of 24 mM of proline increased the endogenous proline level and antioxidant enzyme activity due to cold stress [6] (Table 14).

### 4.6. Proline Effect against Heavy Metal Stress

The exogenous application of proline enhances oil contents, antioxidant enzyme activity, relative water content, organic osmolyte levels, and reduces the hydrogen peroxide levels to combat heavy metal stress. In *Cicer arietinum* (Chickpea), the foliar application of proline improved nitrogen fixation, the nitrogen content in leaves, and antioxidant enzyme activities against cadmium (Cd) metal stress [121]. In *Olea europaea* (Olive), the foliar application of 20 mM of proline enhanced the endogenous proline and oil contents and antioxidant enzyme activities while reducing hydrogen peroxide levels induced by cadmium (Cd) heavy metal exposure [122]. Treatment of *Phaseolus vulgaris* (Bean) with exogenous proline in the culture medium under selenium (Se) heavy metal improved the relative water content, chlorophyll endogenous proline levels, and antioxidant enzyme activities [123]. In *Pisum sativum* (Pea), foliar applications with proline enhanced the growth, photosynthetic activities, relative water content, and organic osmolyte levels when exposed to heavy metals [6]. In *Poncirus trifoliate* (Trifoliate orange), proline supplements in the nutrient solution enhanced protein and cellulose contents despite aluminum (Al) heavy metal exposure [124]. In *Solanum melongena* (Aubergine/Brinjal), seedling treatments with proline increased the endogenous proline content and antioxidant enzyme activities due to arsenate (As) heavy metal stress [125]. In *Triticum aestivum* (Wheat), foliar sprays with 80 mM of proline reduced the ROS and increased the plant height, weight, and photosynthetic capacity compared with untreated plants under the same heavy metal exposure [103]. Exogenous proline applied to *Zea mays* (Maize) improved the defense mechanism and sugar biosynthesis against cadmium (Cd) heavy metal stress [126] (Table 15).

## 5. Glycine Betaine

### 5.1. Structure and Function of Glycine Betaine

Glycine betaine (N, N, N-trimethyl glycine) is an osmoprotectant that also functions as an antioxidant; it is a dipolar molecule that is electrically neutral at its physiological pH. It plays numerous roles in plant growth and metabolism. By preserving the water balance and structural integrity of proteins, it shields cellular components from various stresses. It is now recognized that glycine betaine is synthesized in both the cytosol and chloroplast, and that the production of the chloroplastic form is positively correlated with stress tolerance, whereas that of the cytosolic form is not [127].

The external application of glycine betaine promotes plant growth and protects the plant from harmful substances; it improves stress tolerance by enhancing the relative water content, level of osmolytes, antioxidants and soluble sugars, thus increasing the number of pods, seeds, and leaves per plant, and promoting the overall plant growth, boosting the fresh and dry weights and ultimately the yield of the plant. Glycine betaine also has a protective effect on the oxygen-evolving complex, photosystem II, and plasma membrane by reducing the production of ROS and malondialdehyde. Exogenous glycine betaine protects RuBisCo from oxidative damage and raises carbon fixation and the net photosynthetic rate, protects membrane integrity, and protects subcellular structures through the detoxification of ROS [128].

### 5.2. Glycine Betaine Effect against Drought Stress

The exogenous application of glycine betaine improves crop productivity by increasing antioxidant enzymes activity, soluble sugars, and soluble proteins, improving the osmotic adjustment and photosynthetic rate to combat drought stress. In *Triticum aestivum* (Wheat), the external application of glycine betaine during drought conditions increased the spike length, number of spikelets per spike, number of grains, and leaf turgor potential therefore increasing the total yield; it could have accomplished these effects through an improvement of the stress tolerance index, enhancement in the level of osmolytes, and increase in the relative water content [129,130]. In *Zea mays* (Maize), the foliar application of 100 mM of glycine betaine enhanced the growth, yield, and antioxidant enzyme activities during drought conditions [131]. Similarly in the case of *Solanum lycopersicum* (Tomato), the exogenous application of glycine betaine also improved the yield [132]. Treatment with glycine betaine in *Pisum sativum* (Pea) enhanced the overall growth, numbers of pods and leaves per plant, and the levels of soluble sugars and soluble proteins in the leaves in addition to antioxidant enzyme activities, compared with untreated plants under drought stress conditions [133]. In *Nicotiana tabacum* (Tobacco), the external application of glycine betaine during drought improved plant growth, osmotic adjustment, photosynthesis, and antioxidant enzyme activities [128]. In *Glycine max* (Soybean), the foliar application of glycine betaine at a rate of 3 kg/ha increased the seed number when under drought stress compared with untreated plants [134] (Table 16). 

### 5.3. Glycine Betaine Effect against Salinity Stress

The exogenous application of proline reduces ROS production, lipid peroxidation which increases antioxidant enzymes, soluble sugars, and free amino acids to combat salinity stress. In *Vigna unguiculata* (Cowpea) under salt stress, treatment with glycine betaine increased the soluble sugars and antioxidant enzymes [135]. In *Phaseolus vulgaris* (Common bean), the external application of glycine betaine when grown under saline conditions increased plant fresh weight, leaf area ratio, relative water content, soluble sugars, and free amino acids [136]. In *Oryza sativa* (Rice), the foliar application of glycine betaine increased plant height, fresh and dry weights, chlorophyll content, and reduced malondialdehyde accumulation during salinity stress conditions [128]. Treatment of salt-stressed *Solanum lycopersicum* (Tomato) with glycine betaine increased photosynthesis and stomatal conductance and decreased photorespiration [128]. In *Glycine max* (Soybean), the external application of glycine betaine reduced ROS and lipid peroxidation due to high salinity and increased antioxidant enzyme activities [137]. In *Lolium perenne* (Perennial ryegrass), exogenous glycine betaine was capable of increasing fresh weight and relative water content while reducing electrolyte leakage and malondialdehyde accumulation due to salt stress [138]. The external application of glycine betaine to *Triticum aestivum* (Wheat) growing in high salinity increased the rate of photosynthesis [139] (Table 17).

### 5.4. Glycine Betaine Effect against Heat Stress

The exogenous application of glycine betaine increases the accumulation of heat shock proteins, photosynthetic rate, relative membrane permeability, and ROS scavenging activity to overcome heat stress. In *Solanum lycopersicum* (Tomato) under heat stress, treatment with glycine betaine increased the fruit yield and rate of photosynthesis. The external application of 1 and 5 mM of glycine betaine improved seed germination, increased the expressions of heat shock protein genes, and increased the accumulation of heat shock proteins to alleviate heat stress [128,140]. In *Hordeum vulgare* (Barley), a 10 mM of glycine betaine treatment increased the heat tolerance of photosystem II and had a protective effect on the oxygen-evolving complex. A 10–50 mM glycine betaine application improved the overall growth, photosynthesis, and water relations while decreasing the ion leakage caused by oxidative damage due to high temperatures [141,142]. In *Triticum aestivum* (Wheat), treatment with 100 mM of glycine betaine maintained a higher chlorophyll content, photosystem II photochemical activity, net photosynthetic rate, and resulted in the accumulation of endogenous glycine betaine. The application of 50 and 100 mM of glycine betaine also improved the yield and relative membrane permeability during heat stress [7,85]. *Saccharum officinarum* (Sugarcane) treated with 20 mM of exogenous glycine betaine under heat stress conditions had improved bud sprouting, soluble sugar accumulation, endogenous levels of osmolytes, and decreased hydrogen peroxide production [143]. In *Tagetes erecta* (Marigold), the external application of 0.5 and 1 mM of glycine betaine improved gaseous exchange and reduced ROS accumulation due to heat stress [144] (Table 18).

### 5.5. Glycine Betaine Effect against Cold Stress

The exogenous application of glycine betaine increases photosynthesis, osmolality, prevents chlorosis, and reduces lipid peroxidation to combat cold stress. The foliar application of *Zea mays* (Maize) with the exogenous application of 100 mM of glycine betaine prevented chlorosis and reduced the lipid peroxidation of membranes under cold stress [145]. In *Triticum aestivum* (Wheat), a 100 mM glycine betaine foliar spray increased osmolality and photosynthesis at low temperatures [146]. In *Prunus persica* (Peach), the external application of 10 mM of glycine betaine lowered the malondialdehyde content and increased the endogenous glycine betaine level when the plants were under cold stress [147]. In the case of *Medicago sativa* (Alfalfa) experiencing cold stress, the application of exogenous glycine betaine decreased ion leakage from shoot tissues [148]. In *Solanum lycopersicum* (Tomato), a foliar spray of 0.1 mM of exogenous glycine betaine increased catalase activity, which reduced hydrogen peroxide accumulation as a result of low temperature stress [149]. The external application of glycine betaine also increased the osmolality and endogenous level of glycine betaine in *Hordeum vulgare* (Barley) under cold stress conditions [146] (Table 19).

### 5.6. Glycine Betaine Effect against Heavy Metal Stress

The exogenous application of glycine betaine improves chlorophyll content, antioxidant enzymes activity, reduces lipid peroxidation, and increases gaseous exchange to alleviate heavy metal stress. Spraying 0–100mM of glycine betaine on leaves of *Triticum aestivum* (Wheat) improved the overall growth and chlorophyll, biomass, and protein contents against chromium (Cr) stress [150]. In *Gossypium hirsutum* (Cotton), the application of exogenous glycine betaine (1 mM) improved plant growth, antioxidant enzyme activities, and the rate of photosynthesis and gaseous exchange processes, thus alleviating cadmium (Cd) and lead (Pb) stress [151,152]. A foliar application with 0, 50, and 100 mM of glycine betaine also improved plant growth and alleviated Cr stress in *Vigna radiata* (Mung bean) [153]. In *Amaranthus tricolor* (Amaranth), the external application of glycine betaine improved photosynthesis and the chlorophyll content in leaves and alleviated Cd stress [154]. In *Lolium perenne* (Perennial ryegrass), the application of exogenous glycine betaine improved membrane stability, reduced lipid peroxidation, and alleviated Cd stress [155]. In the case of *Nicotiana tabacum* (Tobacco), treatment with exogenous glycine betaine reduced stomatal closure, malondialdehyde accumulation, and leaf damage, thus alleviating Cd stress [8]. The foliar application of glycine betaine to *Cucumis sativus* (Cucumber) had a significant protective effect on the chlorophyll content of the plant and alleviated aluminum (Al) stress [156]. In *Sorghum bicolor* (Millet), treatment with 50–100 mM of glycine betaine improved plant quality and yield and alleviated Cr stress [157]. In *Oryza sativa* (Rice), the application of exogenous glycine betaine increased the expressions of the glutathione S-transferase (GST) and glutaredoxin (GRX) genes and alleviated arsenic (As) stress [158] (Table 20).

## Figures and Tables

**Figure 1 ijms-23-12913-f001:**
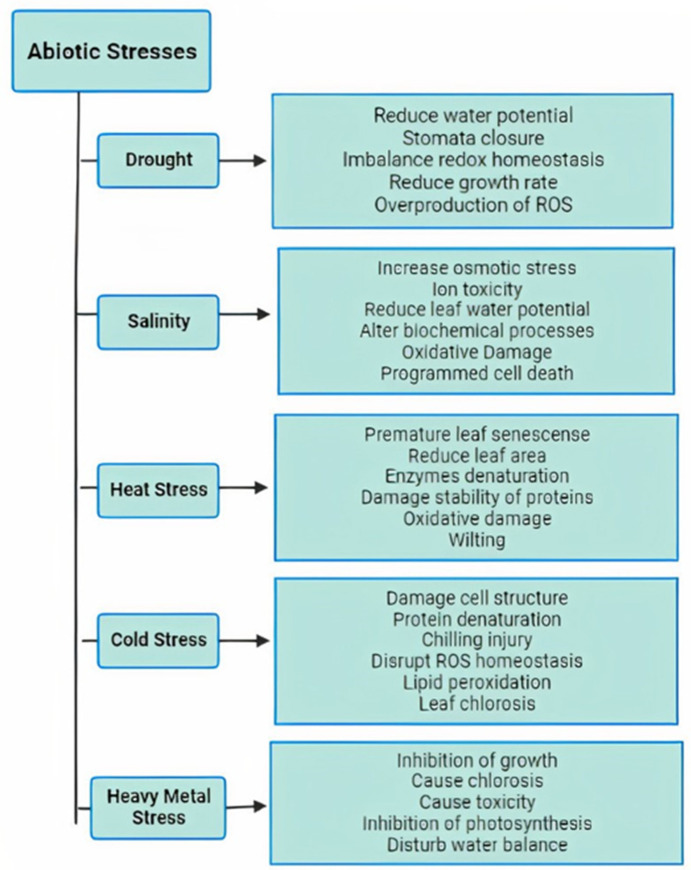
The negative impacts of abiotic stresses on plants.

**Figure 2 ijms-23-12913-f002:**
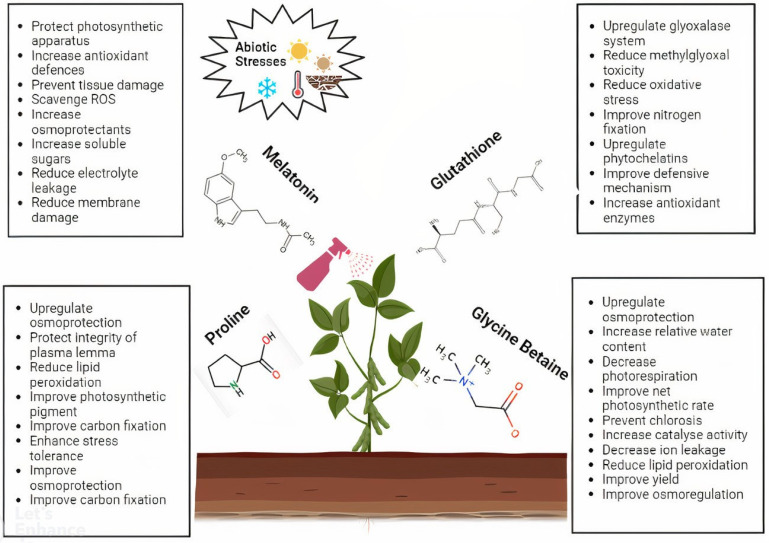
The external application of melatonin, glutathione, proline, and glycine betaine, along with their contribution for increasing the plant’s ability to withstand abiotic stresses.

**Table 1 ijms-23-12913-t001:** Physiological responses of various crops to melatonin treatment against drought stress.

Drought Stress					
Crop	Common Name	Melatonin Dose	Treatment Method	Response to Treatment	Reference(s)
*Arabidopsis thaliana*	Arabidopsis	50 µM	Supplemented with nutrient solution	Upregulated stress-responsive genes and soluble sugars	[14]
*Oryza sativa*	Rice	100 µM	Pretreatment in distilled water for growing	Improved plant growth, osmoprotectants, stress-responsive genes, and ROS scavenging; reduced electrolyte leakage	[15]
*Zea mays*	Maize	1 mM	Supplemented with irrigation	Upregulated photoprotection (photosystem II efficiency)	[16]
*Zea mays*	Maize	100 µM	Foliar application	Increased stomatal conductance, photosynthesis, transpiration rates, cell turgor, and water holding capacity; increased enzymatic and non-enzymatic antioxidants, regulated osmotic potential, and ROS scavenging	[17]
*Triticum aestivum*	Wheat	500 µM	Soil application	Regulated photosynthesis, cell turgor; increased water holding capacity and ROS scavenging; reduced membrane damage	[18]
*Triticum aestivum*	Wheat	10 and 100 µM (dependent on variety)	Seed treatment	Increased germination percentage, radicle and plumule length, and lysine (germination-related amino acid)	[19]
*Fagopyrum tataricum*	Tartary Buckwheat	100 µM	Foliar application	Increased osmoprotectants, water status, secondary metabolites, antioxidant enzymes, photosynthetic rate, and ROS scavenging	[20]
*Hordeum vulgare*	Barley	1 mM	Foliar or soil application	Increased endogenous melatonin, antioxidants, ABA, water status, rate of photosynthesis, and photosystem II efficiency	[21]
*Glycine max*	Soybean	50 µM	Seed coating	Increased seedling biomass and seedling growth; reduced electrolyte leakage	[22]
*Glycine max*	Soybean	100 µM	Foliar and root application	Increased plant growth, flowering, seed yield, gaseous exchange, photosystem II efficiency and antioxidant enzymes	[23]
*Minhot esculenta*	Cassava	100 µM	Soil application	Increased peroxidase activity and ROS scavenging	[24]
*Gossypium hirsutum*	Cotton	100 µM	Seed pre-soaking	Increased number and opening of stomata, antioxidant enzyme activities, osmoprotection, and ROS scavenging	[25]
*Medicago sativa*	Alfalfa	10 µM	Soil application	Increased chlorophyll content, stomatal conductance, and osmoprotection; upregulated nitro-oxidative homeostasis; reduced cellular redox disruption; scavenged ROS	[26]
*Malus domestica*	Apple	100 µM	Soil application	Increased water holding capacity, rate of photosynthesis, stomatal opening regulation, and antioxidants; decreased electrolyte leakage, ROS, oxidative damage, and leaf senescence	[27]
*Vitis vinifer*	Grape	100 µM	Root pretreatment supplemented with irrigation	Increased photoprotection, leaf thickness, stomata size, and enzymatic and non-enzymatic antioxidants; reduced oxidative damage	[4]
*Actinidia chinensis*	Kiwifruit	100 µM	Supplemented with irrigation	Increased osmoprotectants, protein biosynthesis and photosynthesis; reduced cell membrane damage	[12]
*Carya cathayensis*	Chinese hickory	100 µM	Foliar application pretreatment	Increased photosynthesis, antioxidants, and osmoprotectants; scavenged ROS	[13]
*Solanum lycopersicum*	Tomato	0.1 mM	Supplemented with irrigation	Increased photosynthesis, photosystem II efficiency, and antioxidants; reduced toxic substances	[28]
*Solanum lycopersicum*	Tomato	200 µM	Foliar application	Increased chlorophyll and antioxidant enzymes	[29]
*Capsicum annuum*	Pepper	50 µM	Seed pretreatment	Increased water holding capacity, endogenous melatonin, carotenoids, and chlorophyll	[30]
*Citrullus lanatus*	Watermelon	150 µM	Root pretreatment	Increased wax accumulation; reduced abscisic acid	[19]
*Cucumis sativus*	Cucumber	100 µM	Seed priming and nutrient solution	Increased seed germination, root growth, chlorophyll, photosynthesis, antioxidant enzymes, and ROS scavenging	[4]
*Cucumis sativus*	Cucumber	10 µM	Foliar application	Scavenged ROS; improved drought tolerance	[31]
*Brassica napus*	Rapeseed	500 µM	Seed priming	Increased chlorophyll, stomatal regulation, cell wall expansion, antioxidant enzymes, and osmoprotectants; reduced oxidative injury	[32]
*Dendranthema morifolium*	Jinyu Chuju	100 µM	Foliar application	Increased photosynthesis, chlorophyll, and osmoprotectants; reduced cell membrane damage and relative conductivity	[33]
*Dracocephalum moldavica*	Moldavian balm (Dragon head)	100 µM	Foliar application	Increased plant growth and flowering, antioxidant activity, chlorophyll, water holding capacity, and ROS scavenging	[34]
*Agrostis stolonifera*	Creeping bentgrass	20 µM	Foliar application	Increased photosynthetic content, water holding capacity, and photosystem II efficiency; reduced leaf senescence; scavenged ROS	[35]
*Festuca arundinacea*	Tall fescue	20 µM	Irrigation pretreatment	Increased antioxidant enzyme activity, chlorophyll, and plant growth; scavenged ROS	[36]
*Cynodon dactylon*	Bermuda grass	20 and 100 µM	Irrigation pretreatment	Increased plant growth, chlorophyll, antioxidant activity, stress-responsive genes, and hormonal regulation; scavenged ROS	[14]
*Trigonella foenum-graecum*	Fenugreek	100 and 300 µM	Foliar application pretreatment	Increased endogenous melatonin and secondary metabolites, chlorophyll, and antioxidant enzymes; scavenged ROS	[37]
*Coffea arabica*	Coffee	300 µM	Soil application	Increased photoprotection, gaseous exchange, carboxylation activity, chlorophyll, and antioxidant enzyme activities	[38]
*Camellia sinensis*	Tea	100 µM	Foliar application pretreatment	Increased photosynthesis, antioxidant enzymes, and GSH and AsA contents; scavenged ROS	[39]
*Nicotiana benthamiana*	Tobacco	10 µM	Foliar application	Improved drought tolerance; scavenged ROS; reduced oxidative damage	[31]

ROS, reactive oxygen species; GSH, reduced glutathione; AsA, ascorbic acid.

**Table 2 ijms-23-12913-t002:** Physiological responses of various crops to melatonin treatment against salinity stress.

Salinity Stress					
Crop	Common Name	Melatonin Dose	Treatment Method	Response to Treatment	Reference(s)
*Momordica charantia*	Bitter melon	150 µM	Seed priming	Increased relative water content, antioxidant enzyme activities, and gene expression levels; decreased hydrogen peroxide and malondialdehyde levels	[42]
*Zea mays*	Maize	0.4, 0.8, and 1.6 mM	Pretreatment of seeds	Improved shoot and root lengths, germination energy, fresh and dry weights of seedling, K^+^ content, antioxidant enzyme activities, and relative water content.	[43]
*Gossypium hirsutum*	Cotton	25 µM	Seed priming	Enhanced ability to scavenge ROS and improved photosynthetic efficiency	[44]
*Triticum aestivum*	Wheat	70 µM	Seed priming	Enhanced photosynthetic pigments; indole-3-acetic acid content and growth parameters	[45]
*Ocimum basilicum*	Basil	10 µM	Seed priming	Increased contents of flavonoid and phenolic acid	[46]
*Vicia faba*	Faba bean	100 and 500 mM	Seed priming	Improved novel protein expressions	[47]
*Cucumis sativum*	Cucumber	1 µM	Seed priming	Enhanced seed germination	[43]
*Arabidopsis thaliana*	Arabidopsis	10 µM	Foliar application	Induced antioxidant defense system; scavenged ROS; upregulated abscisic acid-responsive genes	[48]
*Brassica napus*	Rapeseed	1 µM	Foliar application	Reduced lipid peroxidation and hydrogen peroxide content; maintained redox and ion homeostasis	[49]
*Brassica juncea*	Mustard greens	1 µM	Foliar application	Increased leaf length and width, plant height, and stem diameter; improved gaseous exchange, relative water content; increased salicylic acid and reduced abscisic acid	[50]
*Cucumis melo*	Melon	0, 10, and 50 µM	Seed pretreatment	Increased seed germination	[51]
*Oryza sativa*	Rice	0, 10, and 20 µM	Root irrigation	Upregulated antioxidants and leaf senescence; inhibited cell death and chlorophyll degradation	[52]
*Glycine max*	Soybean	0–100 µM	Foliar application	Increased photosynthesis, cell division, carbohydrates, fatty acids, and ascorbate contents; reduced inhibitory effect on gene expressions	[22]
*Malus hupehensis*	Pingyitiancha	0.1 µM	Seed pretreatment	Increased photosynthesis and ion homeostasis; reduced oxidative damage	[53]
*Solanum lycopersicum*	Tomato	100 µM	Root irrigation	Increased protein and membrane protection, antioxidant activities, and photosynthesis	[54]
*Citrullus lanatus*	Watermelon	50–150 µM	Seed pretreatment	Increased antioxidant enzymes, photosynthesis, and photosystem II efficiency; reduced stomatal closure and oxidative damage	[55]

ROS, reactive oxygen species.

**Table 3 ijms-23-12913-t003:** Physiological responses of various crops to melatonin treatment against heat stress.

Heat Stress					
Crop	Common Name	Melatonin Dose	Treatment Method	Response to Treatment	Reference(s)
*Triticum aestivum*	Wheat	20 µM	Soil application	Increased rate of photosynthesis; reduced oxidative damage	[56]
*Lolium perenne*	Perennial Ryegrass	100 µM	Foliar application	Regulated cytokinin biosynthesis genes; downregulated abscisic acid biosynthesis genes; enhanced endogenous melatonin level	[57]
*Solanum lycopersicum*	Tomato	100 µM	Seed pretreatment	Enhanced phenolic acid level; regulated transcript abundances; increased endogenous melatonin levels; reduced oxidative stress	[58]
*Arabidopsis thaliana*	Arabidopsis	20 µM	Foliar application	Upregulated heat shock factors	[59]
*Zea mays*	Maize	100 µM	Soil application	Increased photosynthesis; reduced oxidative damage	[17]

**Table 4 ijms-23-12913-t004:** Physiological responses of various crops to melatonin treatment against cold stress.

Cold Stress					
Crop	Common Name	Melatonin Dose	Treatment Method	Response to Treatment	Reference(s)
*Triticum aestivum*	Wheat	100 µM	Spray	Improved antioxidant enzyme activities; reduced oxidative stress	[61]
*Citrullus lanatus*	Watermelon	150 µM	Soil treatment	Increased accumulation of hydrogen peroxide; increased tolerance	[62]
*Solanum lycopersicum*	Tomato	100 µM	Seedling spray	Improved photosynthesis; reduced oxidative damage	[63]
*Cynodon dactylon*	Bermuda grass	100 µM	Foliar application	Increased arabinose, mannose, and propanoic acid levels	[64]
*Hordeum vulgare*	Barley	1 mM	Soil irrigation	Improved water status, antioxidant system, and abscisic acid level	[21]
*Camellia sinensis*	Tea plant	100 µM	Spray	Improved production of antioxidant enzymes; reduced oxidative stress	[65]
*Oryza sativa*	Rice	N/A	Spray	Increased antioxidant enzyme activities; reduced oxidative stress	[55]
*Cucumis sativus*	Cucumber	100 µM	Foliar spray	Improved antioxidant enzyme productions and activities; reduced oxidative stress	[66]

**Table 5 ijms-23-12913-t005:** Physiological responses of various crops to melatonin treatment against heavy metal stress.

Heavy Metal Stress					
Crop	Common Name	Melatonin Dose	Treatment Method	Response to Treatment	Reference(s)
*Triticum aestivum*	Wheat	50 µM	Soil treatment	Increased antioxidant enzymes against cadmium metal stress	[68]
*Medicago sativa*	Alfalfa	50 µM	Foliar application	Increased ABC transporters; decreased cadmium accumulation	[69]
*Solanum lycopersicum*	Tomato	100 µM	Seed priming	Increased antioxidants and plant growth; reduced electrolyte leakage and photoinhibition under cadmium metal stress	[21]
*Nicotiana benthamiana*	Tobacco	15 µM	Foliar application	Increased cell growth and viability; decreased DNA damage against lead heavy metal	[53]
*Cyphomandra betacea*	Tree tomato	50 µM	Soil treatment	Increased antioxidants and plant biomass under cadmium stress	[70]
*Glycine max*	Soybean	100 mM	Seed priming	Increased photosynthesis and antioxidant enzymes under aluminum stress	[53]
*Brassica oleracea*	Red cabbage	10 µM	Foliar application	Increased germination and fresh weight against copper metal stress	[53]
*Citrullus lanatus*	Watermelon	50 mg/L	Seed priming	Increased plant growth, photosynthesis, chlorophyll, antioxidant enzymes, and scavenging of ROS against vanadium metal stress	[71]
*Zea mays*	Maize	500 µM	Soil treatment	Induced additional proteins related to stress reduction during germination	[72]
*Cucumis sativus*	Cucumber	100 and 150 µM	Soil irrigation	Reduced stress-promoted expression of genes e.g., CsHA2 under cadmium metal stress	[43]
*Amaranthus viridis*	Amaranthus	400 µM	Foliar application	Decreased accumulation of metals (e.g., lead) in roots	[73]

ABC transporters, ATP-binding cassette-containing transporters; ROS, reactive oxygen species.

**Table 6 ijms-23-12913-t006:** Physiological responses of various crops to glutathione treatment against drought stress.

Drought Stress					
Crop	Common Name	Glutathione Dose	Treatment Method	Response to Treatment	Reference(s)
*Cicer arietinum*	Chickpea	0.75 mM	Seed soaking	Increased growth parameters, chlorophyll, photosynthesis, endogenous proline, and antioxidant enzyme activities	[74]
*Oryza sativa*	Rice	0.2 mM	Spraying	Increased root and shoot lengths, dry and fresh weights, chlorophyll pigment, relative water content, and antioxidant enzyme activities	[5]
*Brassica napus*	Rapeseed	N/A	Foliar application	Scavenged ROS; reduced oxidative damage	[75]
*Triticum aestivum*	Wheat	N/A	Sprayed	Improved tolerance compared with non-treated cultivar	[76]
*Vigna radiata*	Mung bean	N/A	Exogenous application	Lessened drought-induced oxidative damage through the enhancement of the capacity of the antioxidant system and glyoxalase activity	[77]
*Arabidopsis thaliana*	Arabidopsis	N/A	Spraying	Increased abscisic acid level and tolerance against drought stress; improved plant health under stressful conditions	[78]

ROS, reactive oxygen species; N/A, not available.

**Table 7 ijms-23-12913-t007:** Physiological responses of various crops to glutathione treatment against salinity stress.

Salinity Stress					
Crop	Common Name	Glutathione Dose	Treatment Method	Response to Treatment	Reference(s)
*Capsicum frutescence*	Pepper	0.4 and 0.8 mM	Foliar spray	Increased water use efficiency, growth, fresh and dry weights of roots and shoots, yield, osmoprotectants, and antioxidants	[79]
*Cucumis sativus*	Cucumber	0.5 mM	Seed soaking	Increased growth, fresh and dry weights, relative water content, photosynthetic activity, and stomatal conductance	[80]
*Vicia faba*	Faba bean	0.5 mM	Foliar spray	Increased growth, fresh and dry weights, relative water content, photosynthetic activity, stomatal conductance, and antioxidant enzyme activities	[81]
*Triticum aestivum*	Wheat	1 mM	Foliar spray	Increased plant growth, membrane stability, and accumulation of osmoprotectants	[59]
*Glycine max*	Soybean	1 mM	Foliar spray	Increased growth, photosynthesis, membrane stability, soluble sugars, and antioxidant enzyme activities	[5]
*Phaseolus vulgaris*	Common bean	0.75 mM	Foliar spray	Increased plant length, number and surface area of leaves, fresh and dry weights of plant, relative water content, photosynthesis, and soluble sugars	[5]
*Arabidopsis thaliana*	Arabidopsis	N/A	Foliar application	Increased abscisic acid and tolerance against drought stress; improved plant health under stressful conditions	[78]
*Solanum lycopersicum*	Tomato	N/A	Exogenous application	Improved tolerance against salinity and oxidative stresses; decreased lipid peroxidation	[82]
*Oryza sativa*	Rice	N/A	sprayed	Improved activities of antioxidant enzymes; decreased ROS accumulation and ROS-induced DNA damage	[83]

ROS, reactive oxygen species; N/A, not available.

**Table 8 ijms-23-12913-t008:** Physiological responses of various crops to glutathione treatment against heat stress.

Heat Stress					
Crop	Common Name	Glutathione Dose	Treatment Method	Response to Treatment	Reference(s)
*Triticum aestivum*	Wheat	N/A	External application	Increased antioxidant enzyme activities and resistance to heat stress	[85,87]
*Vigna radiata*	Mung bean	N/A	Seed pretreatment	Increased antioxidant enzyme activities; enhanced stress resistance; decreased ROS level	[77]
*Cucumis sativus*	Cucumber	N/A	External application	Enhanced heat resistance, plant growth, chlorophyll content, and photosynthetic rate	[84]
*Brassica campestris*	Mustard	N/A	External application	Maintained relative water content; increased ROS scavenging and antioxidants	[86]

ROS, reactive oxygen species; N/A, not available.

**Table 9 ijms-23-12913-t009:** Physiological responses of various crops to glutathione treatment against cold stress.

Cold Stress					
Crop	Common Name	Glutathione Dose	Treatment Method	Response to Treatment	Reference(s)
*Oryza sativa*	Rice	0.5 Mm	Spraying	Increased lengths of root and shoot, fresh and dry weights, and endogenous glutathione level	[88]
*Capsicum annum*	Pepper	0.5 Mm	Spraying	Increased lengths of root and shoot, fresh and dry weights, and endogenous glutathione level	[60]
*Cucumis sativus*	Cucumber	N/A	Foliar application	Decreased electrolyte leakage and lipid peroxidation	[89]
*Jatropha curcas*	Purging nut	N/A	External application	Enhanced resistance and antioxidant enzyme activities	[90]

**Table 10 ijms-23-12913-t010:** Physiological responses of various crops to glutathione treatment against heavy metal stress.

Heavy Metal Stress					
Crop	Common Name	Glutathione Dose	Treatment Method	Response to Treatment	Reference(s)
*Triticum aestivum*	Wheat	20 µM	Foliar spray	Increased photosynthetic pigments and endogenous glutathione level against cadmium metal	[62]
*Solanum melongena*	Brinjal(Aubergine)	1 mM	Seed pretreatment	Mitigated adverse effects of stress and protein damage against arsenate metal stress	[91]
*Zea mays*	Maize	30 µM	Foliar application	Increased secondary metabolites and flavonoids; alleviated oxidative damage under cadmium metal stress	[92]
*Lolium multiflorum*	Italian ryegrass	200 µM	External application	Increased stress tolerance and biomass of roots and shoots against lead stress	[93]
*Hordeum vulgare*	Barley	N/A	External application	Improved antioxidant defense system and photosynthesis; decreased ROS accumulation against cadmium metal stress	[94,98]
*Solanum lycopersicum*	Tomato	N/A	External application	Synchronized transcript levels of several stress-responsive transcription factors; improved nitric oxide contents against cadmium metal stress	[95]
*Oryza sativa*	Rice	N/A	External application	Elevated endogenous glutathione level, mineral elements and pigment contents; upregulated phytochelatins; synchronized antioxidant enzyme activities under cadmium metal stress	[96,99]
*Brassica campestris*	Mustard	N/A	Exogenous application	Reduced cadmium levels in roots and leaves and the accumulation of ROS; protected against stress	[97]

ROS, reactive oxygen species; N/A, not available.

**Table 11 ijms-23-12913-t011:** Physiological responses of various crops to proline treatment against drought stress.

Drought Stress					
Crop	Common Name	Proline Dose	Treatment Method	Response to Treatment	Reference(s)
*Zea mays*	Maize	1 mM	Seed priming	Increased photosynthetic and transpiration rates and stomatal conductance	[6]
*Zea mays*	Maize	N/A	Foliar application	Promoted uptake and accumulation of nitrogen, phosphorus, and potassium, as well as tolerance against drought stress	[6]
*Triticum aestivum*	Wheat	150 ppm	Foliar application	Reduced malondialdehyde level and lipid peroxidation	[101]
*Chenopodium quinoa*	Quinoa	N/A	Foliar application	Improved photosynthetic pigments, phenols, free amino acids, plant height, and dry and fresh weights of roots and shoots	[102]
*Arabidopsis thaliana*	Arabidopsis	N/A	Spraying	Scavenged ROS; protected the integrity of plasma lemma	[100]
*Pisum sativum*	Pea	4 mM	Foliar spray	Increased yield, the non-enzymatic antioxidant defense system, and soluble protein concentration	[103]

ROS, reactive oxygen species; N/A, not available.

**Table 12 ijms-23-12913-t012:** Physiological responses of various crops to proline treatment against salinity stress.

Salinity Stress					
Crop	Common Name	Proline Dose	Treatment Method	Response to Treatment	Reference(s)
*Cucumis melo*	Muskmelon	10 mM	Foliar spray	Increased growth, chlorophyll and proline contents, and relative water content	[104]
*Cucumis sativus*	Cucumber	10 mM	Nutrient solution	Increased growth, proline content, and antioxidant enzyme activities	[105]
*Glycine max*	Soybean	25 mM	External application	Increased growth, proline content, antioxidant enzyme activities, and nitrogen fixation	[106]
*Helianthus annus*	Sunflower	30 and 60 mM	Foliar spray	Increased growth, proline and amino acids contents, and antioxidant enzyme activities	[107]
*Zea mays*	Maize	30 mM	Foliar spray	Increased growth and proline content	[108]
*Triticum durum*	Durum wheat	12 mM	Seed pretreatment	Increased growth, photosynthetic activity, proline content, and antioxidant enzyme activities	[109]
*Sorghum bicolor*	Great millet	30 mM	Foliar spray	Increased growth, relative water content, gaseous exchange, and amino acids and proline contents	[110]
*Oryza sativa*	Rice	1, 5, and 10 mM	Seed pretreatment	Increased growth, seed germination, and chlorophyll and proline content	[111]
*Brassica juncea*	Mustard greens	20 mM	Foliar application	Improved yield and stress tolerance	[112]
*Capsicum annum*	Red pepper	0.8 mM	Foliar application	Improved antioxidant enzyme activities, photosynthetic and transpiration rates, plant dry and fresh weights, and root and shoot lengths	[113]
*Daucus carota*	Wild carrot	N/A	Foliar application	Improved antioxidant enzyme activities, and potassium and calcium contents in roots and shoots	[114]
*Phaseolus vulgaris*	Bean	N/A	Foliar application	Improved antioxidant enzyme activities and endogenous proline level	[115]
*Raphanus sativus*	Radish	N/A	Foliar application	Improved transpiration rate, stomatal conductance, pigment contents, and levels of proteins and some nutrients	[116]
*Vicia faba*	Faba bean	N/A	Foliar application	Improved levels of photosynthetic pigments, soluble carbohydrates, endogenous proline, and free amino acids	[6]
*Lactuca sativa*	Lettuce	5 µM	Foliar application	Improved plant growth, photosynthetic rate, chlorophyll content and yield	[117]

**Table 13 ijms-23-12913-t013:** Physiological responses of various crops to proline treatment against heat stress.

Heat Stress					
Crop	Common Name	Proline Dose	Treatment Method	Response to Treatment	Reference(s)
*Abelmoschus esculentus*	Okra	N/A	Foliar application	Improved shoot length, number of leaves per plant, and free amino acids content	[118]
*Lactuca sativa*	Lettuce	5µM	Foliar application	Improved plant growth, photosynthetic rate, chlorophyll content and yield	[117]
*Vigna radiata*	Mung bean	N/A	Foliar application	Improved carbon dioxide (CO_2_) assimilation capacity and heat tolerance	[119]

**Table 14 ijms-23-12913-t014:** Physiological responses of various crops to proline treatment against cold stress.

Cold Stress					
Crop	Common Name	Proline Dose	Treatment Method	Response to Treatment	Reference(s)
*Citrus reticulata*	Mandarin orange	N/A	Foliar application	Increased contents of phenolic acids, flavonoids and endogenous proline; increased antioxidant enzyme activities	[120]
*Citrus sinensis*	Sweet orange	N/A	Foliar application	Increased contents of phenolic acids, flavonoids, and endogenous proline; increased antioxidant enzyme activity	[120]
*Citrus paradisi*	Grapefruit	N/A	Foliar application	Increased contents of phenolic acids, flavonoids and endogenous proline; increased antioxidant enzyme activity	[120]
*Capsicum annum*	Red pepper	24 mM	Foliar application	Increased endogenous proline level and antioxidant enzyme activities	[6]

**Table 15 ijms-23-12913-t015:** Physiological responses of various crops to proline treatment against heavy metal stress.

Heavy Metal Stress					
Crop	Common Name	Proline Dose	Treatment Method	Response to Treatment	Reference(s)
*Cicer arietinum*	Chickpea	N/A	Foliar application	Improved nitrogen fixation, nitrogen content in leaves, and antioxidant enzyme activities against cadmium stress	[121]
*Olea europaea*	Olive	20 mM	Irrigation	Enhanced proline and oil contents, antioxidant enzyme activities; reduced hydrogen peroxide against cadmium stress	[122]
*Phaseolus vulgaris*	Bean	N/A	Culture medium	Improved relative water content, chlorophyll and endogenous proline contents, and antioxidant enzyme activities under selenium heavy metal stress	[6,123]
*Pisum sativum*	Pea	N/A	Foliar application	Enhanced growth, photosynthetic activity, relative water content and organic osmolyte contents	[6]
*Poncirus trifoliata*	Trifoliate orange	N/A	Nutrient solution	Enhanced protein and cellulose contents against aluminum stress	[124]
*Solanum melongena*	Aubergine	N/A	Seedling treatment	Increased endogenous proline level and antioxidant enzyme activities under arsenate stress	[6,125]
*Triticum aestivum*	Wheat	80 mM	Foliar spray	Reduced ROS; increased plant height, weight, and photosynthetic capacity	[103]
*Zea mays*	Maize	N/A	Exogenous application	Improved defensive mechanism and sugar biosynthesis against cadmium stress	[126]

**Table 16 ijms-23-12913-t016:** Physiological responses of various crops to glycine betaine treatment against drought stress.

Drought Stress					
Crop	Common Name	Glycine Betaine Dose	Treatment Method	Response to Treatment	Reference(s)
*Triticum aestivum*	Wheat	N/A	Exogenous application	Improved stress tolerance index; enhanced osmolyte and relative water contents	[129]
*Triticum aestivum*	Wheat	N/A	Exogenous application	Increased spike length, number of spikelets per spike, number of grains, yield, and leaf turgor potential	[130]
*Zea mays*	Maize	100 mM	Foliar application	Enhanced growth, yield, and antioxidant enzyme activities	[131]
*Solanum lycopersicum*	Tomato	N/A	Exogenous application	Improved yield	[132]
*Pisum sativum*	Pea	N/A	Exogenous application	Enhanced growth, number of pods and leaves per plant; increased level of soluble sugars and soluble protein in leaves; increased antioxidant enzyme activities	[133]
*Nicotiana tabacum*	Tobacco	80 mM	Foliar application	Improved plant growth, osmotic adjustment, photosynthesis, and antioxidant enzyme activities	[128]
*Glycine max*	Soybean	3 kg/ha	Exogenous application	Increased seed number	[134]

**Table 17 ijms-23-12913-t017:** Physiological responses of various crops to glycine betaine treatment against salinity stress.

Salinity Stress					
Crop	Common Name	Glycine Betaine Dose	Treatment Method	Response to Treatment	Reference(s)
*Vigna unguiculata*	Cowpeas	5–10 mM	Foliar application	Increased soluble sugar contents and antioxidant enzymes	[135]
*Phaseolus vulgaris*	Common bean	N/A	Exogenous application	Increased plant fresh weight, leaf area ratio, relative water content, and soluble sugar and free amino acid contents	[136]
*Oryza sativa*	Rice	N/A	Foliar application	Increased plant height, fresh and dry weights, and chlorophyll content; reduced malondialdehyde content	[128]
*Solanum lycopersicum*	Tomato	N/A	Exogenous application	Increased photosynthesis and stomatal conductance; decreased photorespiration	[128]
*Glycine max*	Soybean	N/A	Exogenous application	Reduced ROS and lipid peroxidation; increased antioxidant enzyme activities	[137]
*Lolium perenne*	Perennial ryegrass	0, 20, 50 mM	Exogenous application	Increased fresh weight and relative water content; reduced electrolyte leakage and malondialdehyde content	[138]
*Triticum aestivum*	Wheat	N/A	Exogenous application	Increased rate of photosynthesis	[139]

ROS, reactive oxygen species; N/A, not available.

**Table 18 ijms-23-12913-t018:** Physiological responses of various crops to glycine betaine treatment against heat stress.

Heat Stress					
Crop	Common Name	Glycine Betaine Dose	Treatment Method	Response to Treatment	Reference(s)
*Solanum lycopersicum*	Tomato	N/A	External application	Increased fruit yield and rate of photosynthesis	[128]
*Solanum lycopersicum*	Tomato	1, 5 mM	External application	Improved seed germination, expression of heat shock genes, and accumulation of heat shock proteins	[140]
*Hordeum vulgare*	Barley	N/A	External application	Increased tolerance of photosystem Ⅱ; protective effect on oxygen-evolving complex	[141]
*Hordeum vulgare*	Barley	10 mM	External application	protective effect on oxygen-evolving complex; greater photosystem Ⅱ stability	[141]
*Hordeum vulgare*	Barley	10, 20, 30, 40 and 50 mM	External application	Improved growth, photosynthesis, and water relations; decreased ion leakage	[142]
*Triticum aestivum*	Wheat	100 mM	External application	Maintained higher chlorophyll content, photosystem Ⅱ photochemical activity, net photosynthetic rate, and accumulation of endogenous glycine betaine	[85]
*Triticum aestivum*	Wheat	50 and 100 mM	External application	Improved yield and relative membrane permeability	[7]
*Saccharum officinarum*	Sugarcane	20 mM	External application	Improved bud sprouting, soluble sugar accumulation, and endogenous level of osmolytes; decreased hydrogen peroxide	[143]
*Tagetes erecta*	Marigold	0.5 and 1 mM	External application	Improved gaseous exchange; reduced ROS accumulation	[144]

ROS, reactive oxygen species. N/A: information not available.

**Table 19 ijms-23-12913-t019:** Physiological responses of various crops to glycine betaine treatment against cold stress.

Cold Stress					
Crop	Common Name	Glycine Betaine Dose	Treatment Method	Response to Treatment	Reference(s)
*Zea mays*	Maize	100 mM	Foliar application	Prevented chlorosis; reduced lipid peroxidation of membrane	[145]
*Triticum aestivum*	Wheat	100 mM	Foliar spray	Increased osmolality and photosynthesis	[146]
*Prunus persica*	Peach	10 mM	External application	Lowered malondialdehyde content; increased endogenous glycine betaine level	[147]
*Medicago sativa*	Alfalfa	0.2 M	Seedling sprayed	Decreased ion leakage from shoot tissues	[148]
*Solanum lycopersicum*	Tomato	0.1 mM	Foliar spray	Increased catalase activity; reduced hydrogen peroxide	[149]
*Hordeum vulgare*	Barley	N/A	External application	Increased osmolality and endogenous level of glycine betaine	[146]

**Table 20 ijms-23-12913-t020:** Physiological responses of various crops to glycine betaine treatment against heavy metal stress.

Heavy Metal Stress					
Crop	Common Name	Glycine Betaine Dose	Treatment Method	Response to Treatment	Reference(s)
*Triticum aestivum*	Wheat	0–100 mM	Spraying on leaves	Improved growth, chlorophyll contents, and biomass and protein contents against chromium stress	[150]
*Gossypium hirsutum*	Cotton	1 mM	Exogenous application	Improved plant growth, antioxidant enzyme activities and photosynthetic rate and gaseous exchange; alleviated cadmium stress	[151]
*Gossypium hirsutum*	Cotton	N/A	Foliar application	Improved plant growth and gas attributes; alleviated lead stress	[152]
*Vigna radiata*	Mung bean	0, 50, 100 mM	Foliar application	Improved plant growth; alleviated chromium stress	[153]
*Amaranthus tricolor*	Amaranth	N/A	Exogenous application	Improved photosynthesis and chlorophyll content of leaves; alleviated cadmium stress	[154]
*Lolium perenne*	Perennial ryegrass	N/A	Exogenous application	Improved membrane stability; reduced lipid peroxidation; alleviated cadmium stress	[155]
*Nicotiana tabacum*	Tobacco	N/A	Exogenous application	Reduced stomatal closure, accumulation of malondialdehyde, and leaf damage; alleviated cadmium stress	[8]
*Cucumis sativus*	Cucumber	N/A	Foliar application	Significant protective effect on chlorophyll content; alleviated aluminum stress	[156]
*Sorghum bicolor*	Millet	50–100 mM	Exogenous application	Improved quality and yield; alleviated chromium stress	[157]
*Oryza sativa*	Asian rice	N/A	Exogenous application	Increased GST and GRX gene expressions; alleviated arsenic stress	[158]

GST, glutathione S-transferase; GRX, glutaredoxin; N/A, not available.

## Data Availability

Not applicable.

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
