# Peer review of "Using Exogenous Melatonin, Glutathione, Proline, and Glycine Betaine Treatments to Combat Abiotic Stresses in Crops"

_ijms, 2022, doi:10.3390/ijms232112913_

Round 1
Reviewer 1 Report
the paper entitled 'Using Exogenous Melatonin, Glutathione, Proline and Glycine Betaine Treatments to Combat Abiotic Stresses in Crops' by Khalid et al. is dealing with an imprtant topic. Nevertheless, I have some recommendations and comments (attached in the PDF) to improve the quality of this content.
1/Abstract: this section should be improved by inserting some conclusions and constatation
2/Introduction: This section should clearly state what has been achieved and what will be added to the part of knowledge regarding the topic of the paper
3/ The rest of the paper should consider the comments and mistakes highlighted directly in the attached file while editing the manuscript and try in every section to give explanation and conclusion to the treated part of bibliographical research by the authors
I recommend also the author (If possible of course) to think about a classification according to the dose of treatment for glutathione, proline etc or by crop type of course if author notuced such a variation and dependence.
I have attached many comments attached

Author Response
Comment#1 Abstract: this section should be improved by inserting some conclusions and constatation
Answer#1 Conclusions added as suggested (lines 27-30).
Comment#2 Introduction: This section should clearly state what has been achieved and what will be added to the part of knowledge regarding the topic of the paper
Answer#2 Revised accordingly (lines 79-83).
Comment#3 The rest of the paper should consider the comments and mistakes highlighted directly in the attached file while editing the manuscript and try in every section to give explanation and conclusion to the treated part of bibliographical research by the authors
Answer#3 Revised accordingly (highlighted in the revised manuscript).
Comment#4 I recommend also the author (If possible of course) to think about a classification according to the dose of treatment for glutathione, proline etc or by crop type of course if author notuced such a variation and dependence.
I have attached many comments attached
Answer#4 Revised accordingly as much as possible (highlighted in the revised manuscript).
Reviewer 2 Report
The review of “Using Exogenous Melatonin, Glutathione, Proline and Glycine Betaine Treatments to Combat Abiotic Stresses in Crops” well summarized the various protection effects and related applications of four small-molecular antioxidants against abiotic stresses (drought, salinity, heat, cold, and heavy metal stress). The manuscript was well prepared except some minimum issues.
1. lines 11-28, the abstract may not just simply list various protection effects of four compounds. A more compact and comprehensive viewpoint extracted from the manuscript is recommended for the content of abstract, which can be more enlightened and attractive for the readerships.
2. Lines 78-79,
(1) the resolution of two figures here is too low. The clearer (or original) ones may be used to replace them.
(2) what is the caption or title for the figure in page 3?
3. Please unify the color and clean the revision records in the main text. The font size in the tables may be adjusted for some unfitted content, such as tables 8, 12 and 18.
4. The format of references may be unified. (keeping DOI or not?)
Author Response
Comment#1 lines 11-28, the abstract may not just simply list various protection effects of four compounds. A more compact and comprehensive viewpoint extracted from the manuscript is recommended for the content of abstract, which can be more enlightened and attractive for the readerships.
Answer#1 Conclusion are added accordingly (lines 27-30).
Comment#2 Lines 78-79,
- the resolution of two figures here is too low. The clearer (or original) ones may be used to replace them.
Improved accordingly.
- what is the caption or title for the figure in page 3?
Title added (line 88).
Comment#3 Please unify the color and clean the revision records in the main text. The font size in the tables may be adjusted for some unfitted content, such as tables 8, 12 and 18.
Answer#3 Done accordingly.
Comment#4 The format of references may be unified. ( keeping DOI or not?)
Answer#4 Done accordingly.
Round 2
Reviewer 1 Report
I have noticed a susbstantial revision made to the manuscript
I recommend the publication of the manuscript at this stage.